# Entropy-Based Time Window Features Extraction for Machine Learning to Predict Acute Kidney Injury in ICU

Chun-Te Huang [1], Rong-Ching Chang [2], Yi-Lu Tsai [3], Kai-Chih Pai [2,3], Tsai-Jung Wang [1], Chia-Tien Hsu [1], Cheng-Hsu Chen [1], Chien-Chung Huang [2,4], Min-Shian Wang [4], Lun-Chi Chen [2,3], Ruey-Kai Sheu [2,3], Chieh-Liang Wu [1] and Chun-Ming Lai [2,3,*]

[1] Nephrology and Critical Care Medicine, Department of Internal Medicine and Critical Care Medicine, Taichung Veterans General Hospital, Taichung 407, Taiwan; chunte@vghtc.gov.tw (C.-T.H.); tjwang@vghtc.gov.tw (T.-J.W.); jatenhsu@gmail.com (C.-T.H.); cschen@vghtc.gov.tw (C.-H.C.); clwu@vghtc.gov.tw (C.-L.W.)

[2] Department of Computer Science, Tunghai University, Taichung 407, Taiwan; g08350003@thu.edu.tw (R.-C.C.); kcpai@thu.edu.tw (K.-C.P.); ccwhuang@vgtc.gov.tw (C.-C.H.); lunchi@thu.edu.tw (L.-C.C.); rickysheu@thu.edu.tw (R.-K.S.)

[3] DDS-THU AI Center, Tunghai University, Taichung 407, Taiwan; yltsai@thu.edu.tw

[4] Computer Center, Artificial Intelligence Studio, Taichung Veterans General Hospital, Taichung 407, Taiwan; minnshyan@vghtc.gov.tw

* Correspondence: cmlai@thu.edu.tw

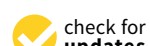



**Featured Application: The proposed method has been applied and landed in the ICU at Taichung Veterans General Hospital since 2021.**

**Abstract:** Acute kidney injury (AKI) refers to rapid decline of kidney function and is manifested by decreasing urine output or abnormal blood test (elevated serum creatinine). Electronic health records (EHRs) is fundamental for clinicians and machine learning algorithms to predict the clinical outcome of patients in the Intensive Care Unit (ICU). Early prediction of AKI could automatically warn the clinicians to review the possible risk factors and act in advance to prevent it. However, the enormous amount of patient data usually consists of a relatively incomplete data set and is very challenging for supervised machine learning process. In this paper, we propose an entropy-based feature engineering framework for vital signs based on their frequency of records. In particular, we address the missing at random (MAR) and missing not at random (MNAR) types of missing data according to different clinical scenarios. Regarding its applicability, we applied it to establish a prediction model for future AKI in ICU patients using 4278 ICU admissions from a tertiary hospital. Our result shows that the proposed entropy-based features are feasible to be used in the AKI prediction model and its performance improves as the data availability increases. In addition, we study the performance of AKI prediction model by comparing different time gaps and feature windows with the proposed vital sign entropy features. This work could be used as a guidance for feature windows selection and missing data processing during the development of a prediction model in ICU.

**Keywords:** acute kidney injury (AKI); machine learning; entropy

## 1. Introduction

Acute kidney injury (AKI) is a medical term to describe rapid decline of kidney function within seven days. The most broadly accepted definition of AKI is proposed by Kidney Disease Improving Global Outcomes (KDIGO) [1] using serum creatinine, a waste product of muscle considered to be an endogenous filtration marker to assess renal function, and urine output to diagnose and define the severity of AKI. The prevalence of AKI is estimated to be around 55–60% in intensive care units (ICU), and is associated with an increased risk of prolonged hospital stay, renal replacement therapy, mortality,

and high medical cost [2]. Early prediction of AKI could help clinicians to provide timely intervention and probably avoid the grave prognosis of end-stage renal disease.

With the growth of computing power, artificial intelligence has been applied extensively in various fields from fingerprint recognition to arrhythmia prediction [3–5]. The clear and objective definition of AKI provide an ideal labelling outcome for machine learning. Tomašev et al. [6] developed a deep learning approach for continuous prediction of future AKI using more than 700,000 patients Electronic Health Records (EHRs) from a multi-site dataset of US Department of Veterans Affairs. Based on their prediction model, clinicians could be warned 48 h in advance before AKI occurrence. However, the prediction algorithm created by Tomašev et al. consists of more than 100 features and may not be easily applied to other hospitals where many features might have high proportion of missing data.

Entropy, also known as Shannon's Entropy or self-information of an event, denotes the uncertainty and contributed information about the state of the system or data [7]. Entropy can be interpreted as a measure of the probability distribution for the amount of missing information [8]. The occurrence of the random event has higher information contribution than the common event observed. Entropy has been applied in numerous research fields, including medical, biometrics, and other real-world applications. Li et al. [9] applied entropy analysis to Electroencephalography (EEG) data to examine its performance on epilepsy detection. Chicote et al. [10] used fuzzy entropy and sample entropy as predictors for Out-of-hospital cardiac arrest (OHCA). Their study showed that entropy is a reliable predictor and outperformed other predictors. Chen et al. [11] concluded that multi-scale entropy could potentially be an early predictor of stroke-in-evolution in ICU-admitted non-atrial fibrillation stroke patients.

In this study, we use Random Forest (RF) machine learning algorithm to build the AKI prediction model [12]. RF has become one of the major machine learning methods, due to its robustness and non-linear ensemble nature and has been used in ICU early warning system [13,14]. RF algorithm can use cross-entropy and entropy-as-loss-functions to find the best split in a classification tree. However, directly applying it without considering the potential impact of missing data may result in poor model performance or bias. In order to conquer the challenges of missing data in EHRs, we take the advantages of entropy in developing early prediction model for AKI, and focus our research question on which data in which time windows should be included to build AKI prediction model. We propose a novel entropy-based feature engineering framework for vital signs and navigate the research question through finding an appropriate feature window for clinical feature inclusion and time gap for subsequent AKI prediction.

The rest of this paper is organized as follows: Section 2 describes the methodology, research material, and the proposed entropy-based feature engineering framework. Section 3 presents the machine learning experimental results using the proposed entropy-based feature engineering framework, time gap variations, and feature window variations. Section 4 provides a discussion and details the limitations of the proposed framework. Section 5 concludes this study.

## 2. Materials and Methods

In this section, we explain the data source, problem definition, and the proposed entropy-based feature engineering framework.

### 2.1. Study Population

Taichung Veterans General Hospital (TVGH) is a tertiary teaching hospital with 112 adult ICU beds in central part of Taiwan. 24,518 adult ICU admissions between July 2015 and December 2019 in the TVGH EHRs database were extracted for analysis. The inclusion criteria is age over 20 years old adults. Exclusion criteria are (1) End stage renal disease with the International Classification of Diseases Tenth Revision (ICD-10) code assignment of N18.6. (2) Had been treated with renal replacement therapy before the

index ICU admission. (3) Received renal replacement therapy within 24 h after index ICU admission. (4) Stayed less than 24 h in ICU. (5) Had an invalid or missing urine output (UO) record. After selection, 15,702 adult ICU admissions were included for final analysis.

AKI was labelled according to Section 2.2, where 9589 AKI patients and 6067 non-AKI patients were identified, respectively. To conduct research up to 72 h before AKI occurrence, we set additional filtering criteria for both AKI patients and non-AKI patients to ensure EHRs are longer than 72 h of time frame. The advantage of such filtering is that it gave us more data and longer period to study patient conditions. Additionally, by choosing a minimal 72-h time frame in ICU for both AKI and non-AKI group, we alleviate the possible bias in non-AKI group selection. Celi et al. [15] performed AKI mortality prediction on MIMIC III data with the Simplified Acute Physiology Score (SAPS), also focused on patients who survived in ICU for more than 72 h using multi-variable regression models.

We obtained a final set of 4278 patients from both groups after filtering with greater than or equal to 72 h admission time, containing 1631 AKI patients and 2647 non-AKI patients. Figure 1 shows the data cohort workflow.

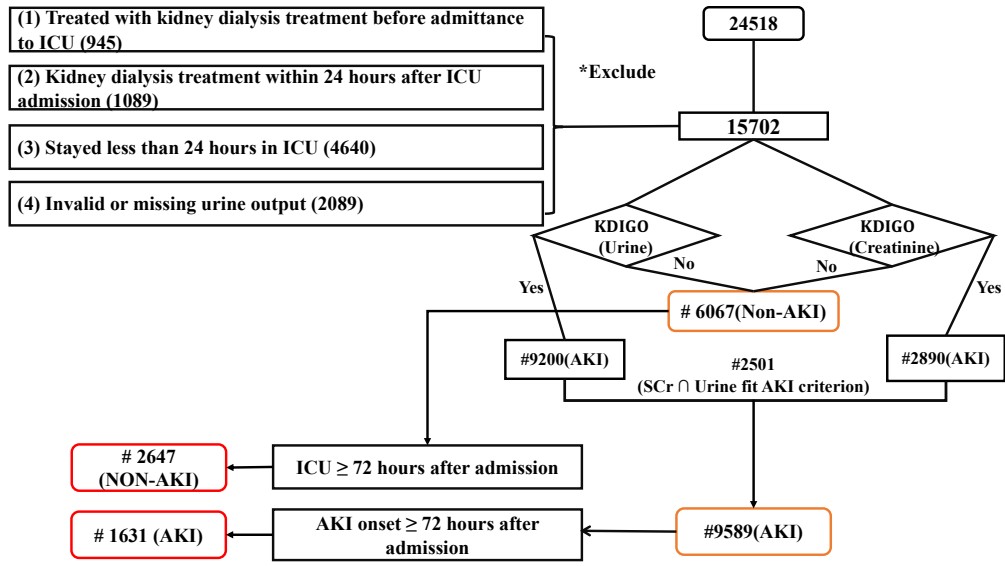

**Figure 1.** Data cohort workflow.

## 2.2. AKI Definition

AKI was defined, identified, and labeled according to the internationally accepted 'Kidney Disease: Improving Global Outcomes' (KDIGO) criteria [16], using both serum creatinine (SCr) and UO under three conditions: (1) SCr increased by 0.3 mg/dl within 48 h; (2) SCr increased more than 1.5 times baseline within 7 days; and (3) UO was less than 0.5 mL/kg/h over 6 h.

## 2.3. Problem Formulation

For this study, we formulated the primary objective to be the early prediction of AKI as a binary classification task. Patients who developed AKI according to the KDIGO criteria were labeled as the positive class, while patients who did not develop AKI were labeled as the negative class. Samples were patient EHR records, with a series of vital sign values with time stamps. Each patient had a unique patient ID, $i$, and unique admission ID, $j$, for every ICU admission.

Formally, $y = f(F)$ was our target function. Our goal in early AKI prediction was to learn a function $f(F)$ that maps a set of features $F$ to predict the binary outcome of $y \in \{-1, 1\}$, where $y$ is equal to 1 if the patient has a positive AKI label and $y$ with non-AKI labels is indicated by $-1$. Patient $i$ was admitted to the ICU with $j$ as their unique admission ID, while $t$ denotes the timestamps. Time gap is the time between AKI onset



and how many hours in advance before the onset do we want to predict the onset event without available data during the gap but using the data from feature window instead. In our study, we denote time gap as $T_{TP} = [0, 48]$ h before the onset. Feature window $T_{FW} = [24, 48]$ is the number of hours before AKI onset, and we collect data with in this time frame. The collected data during this $T_{FW}$ time is being used to predict if the patient will have AKI onset.

Our main objective was to study the relationship between AKI onset prediction with different time gaps and feature windows using the vital sign feature engineering framework we developed. Specifically, we considered the following two time ranges:

1.  Time gap $T_{TP}$ variation: The time gap from AKI onset to 48 h $T_{TP} = [0, 48]$ before onset, using a feature window $T_{FW} = 24$ of 24 h. In another word, we conducted a research where the time gap $T_{TP}$ rolling from 0 h before onset to 48 h before onset. Coupled with.
2.  Feature window $T_{FW}$ variation: The feature window size from 24 h to 48 h before onset, with a time gap of 24 h.

Figure 2 illustrates the AKI prediction task, with a sequential EHRs problem set-up, using a feature window of 24 h versus 48 h with a time gap of 24 h. Non-AKI patients with a feature window of 24 h were randomly selected [17]. Table 1 shows the symbols used in this study.

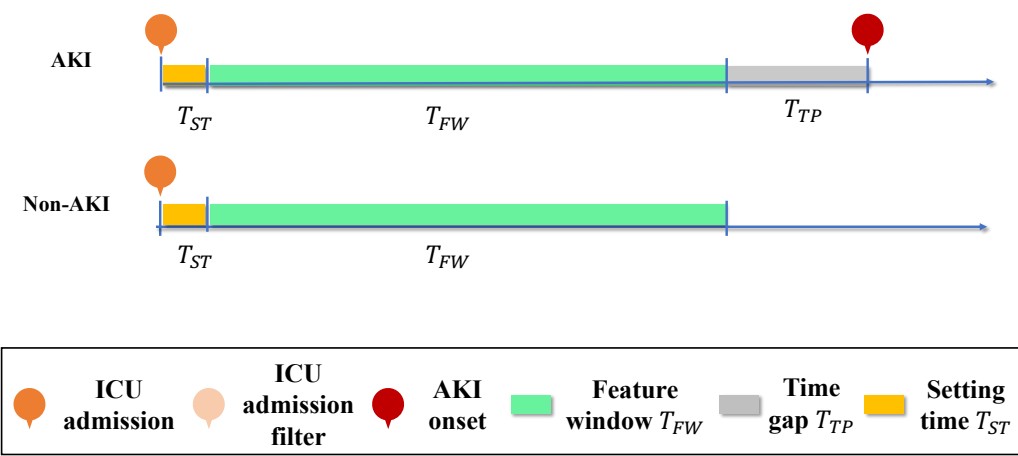

**Figure 2.** Representation of the EHRs and AKI prediction task. AKI onset prediction with feature window variation of 24 h and 48 h with a time gap of 24 h is illustrated. Non-AKI patients with a feature window of 24 h were randomly selected.

### 2.4. Input Feature

Input features were broken down into two parts: Vital signs and administration information. Let $F = (I, J, T, V)$ be a set of features defined by a set of $n$ patients $I = \{i_1, ..., i_n\}$, a set of $m$ admission IDs $J = \{j_1, ..., j_m\}$, a set of time stamps $T$, and a set of $k$ vital signs $V_{k,t} \in V$ at time $t$.

Vital Signs were recorded every 2 h, including systolic blood pressure (SBP), diastolic blood pressure (DBP), pulse pressure, oximetry, respiratory rate, pulse rate, and body temperature. Thus, we had 7 vital sign input features in total. Each feature included 3 generated features, as detailed in Section 2.5.2, and so $k$ was 21 in our study.

**Table 1.** Notation.

| Notations | Definition |
|---|---|
| $y = f(F)$ | Target function |
| $F = (I, J, T, V, L)$ | Set of features |
| $I$ | Set of patients $I = \{i_1, ..., i_n\}$ |
| $J$ | Set of admission ID $J = \{j_1, ..., j_m\}$ |
| $T$ | Set of time stamps |
| $T_{TP}$ | Time gap where $T_{TP} = [0, 48]$ h before AKI onset |
| $T_{ST}$ | Setting time |
| $T_{FW}$ | Feature window where $T_F W = [24, 48]$ h before AKI onset |
| $V$ | Set of vital signs $V_{k,t} \in V$ |
| $n$ | Total number of patients |
| $m$ | Total number of admission IDs |
| $k$ | Total number of vital signs |
| $t$ | Time stamp |
| $H_q(V)$ | Entropy for Vital sign V at the $q^{th}$ time |
| $H$ | Target entropy |
| $h$ | Total number of possible states |
| $D0$ | AKI onset |
| $D1$ | 24 h before AKI onset |
| $D2$ | 48 h before AKI onset |
| $D3$ | 72 h before AKI onset |
| $V_{general}$ | Mean and variance of vital signs |
| $V_{entropy}$ | Entropy of vital signs |
| $V_{merge}$ | The combination of both $V_{general}$ and $V_{entropy}$ |
| $V_{mean}$ | Mean of vital signs |
| $V_{variance}$ | Variance of vital signs |
| $fw$ | Feature window length |
| $q$ | q is an iterative counter denoting the $q^{th}$ time, where $q \in [1, 150]$ |
| $N$ | Number of computation times |

*2.5. Entropy-Based Feature Engineering Framework*

Model performance strongly correlates with data quality, where missing data may result in poor performance. Hence, we proposed a novel entropy-based feature engineering framework which is suitable for vital signs, based on their frequency of occurrence considering clinical availability. This framework was constructed through the following steps:

1. Step 1: We evaluated the setting time $T_{ST}$, which is the time interval between the patient's first admission time and their first data entry. This step is important, as the setting time dramatically affects the portion of missing data. In another word, we only consider data collected after setting time $T_{ST}$.

2. Step 2: The Shannon entropy was used to evaluate all vital signs $V$. This measures the probability distribution that characterizes the amount of missing information and data quality.

3. Step 3: We conducted missing value imputation on $V$. Both Steps 1 and 2 are critical for the data quality, as it is not measured on a frequent and consistent basis; yet, vital signs are crucial for AKI evaluation and indication. An overall workflow is shown in Figure 3.

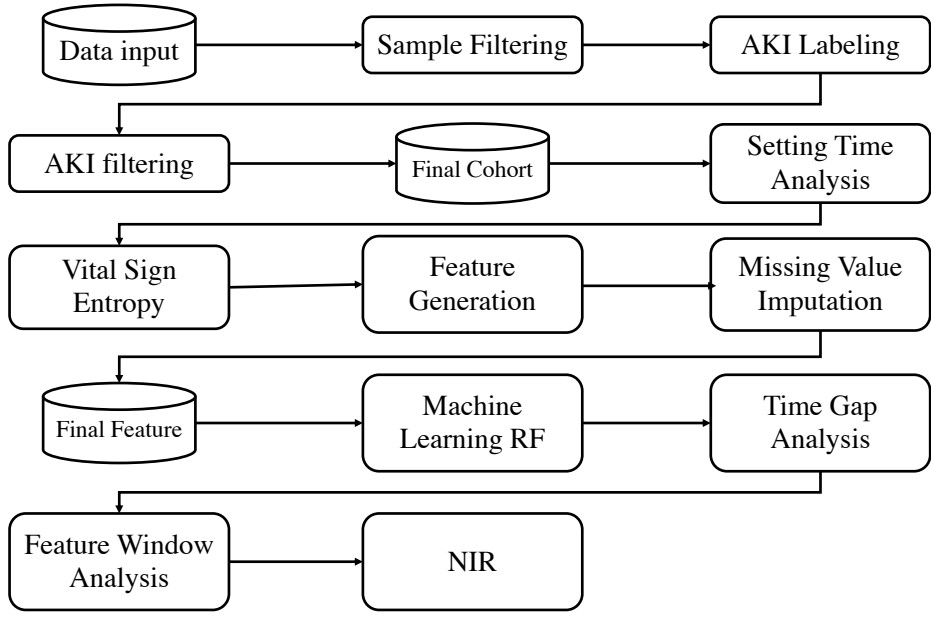

**Figure 3.** Overall framework.

### 2.5.1. Setting Time

The setting time, $T_{ST}$, is defined as the time interval between a patient $i$'s EHRs registered admission time with admission ID $j$ and their first vital sign data entry. This evaluation is critical, with a direct contribution to the amount of missing data and, thus, affecting the model performance. The setting time analysis results are given in Section 3.2. In this step, we are emphasizing the Missing not at random (MNAR) type of missing data that was being introduced by the practical clinical administration setup.

### 2.5.2. Vital Sign Entropy Feature Generation

Consider a discrete vital sign variable $V$ for vital sign feature $k$, which occurs with probability $P(V_{k,1}), ..., P(V_{k,h})$, where $h$ is the total number of possible states. According to the Shannon entropy, defined in [7], the entropy $H$ of the discrete variable $V$ at the $q^{\text{th}}$ time($q \in [1, 150]$) can be written as follows:

$$H_q(V) = -\sum_{q=0}^{h} P(V_q) log P(V_q), \tag{1}$$

where $fw$ is the feature window length. Then, $h$ is:

$$h = \frac{|fw|}{4}. \tag{2}$$

We propose our entropy-based vital sign feature, $H$, as follows:

$$H = \frac{1}{N} \sum_{q=0}^{N} H_q(V), \quad \text{where} \quad N = 150. \tag{3}$$

In our setting, we set $h$ equal to the feature window divided by 4, such that $h = [6, 12]$. As our feature window started at 24 h, $h$ started at 6 and was incremented to 12 as the feature window became longer. Let $N$ be the number of computation times. We obtained the numbers of 4 and $N$ from multiple trials. Our proposed entropy feature, from Equation (3), takes $h$ data points from the vital signs $V$ to calculate $q$ times of entropy, $H_q$, which are then accumulated and normalized $N$ times to obtain our final entropy, $H$. The benefit of this is

that we obtain an entropy that objectively represents the overall data. The entropy can be regarded as another measure of the variation given feature windows.

The calculation of entropy on all vital signs $V$ provides a quantified information evaluation of the average uncertainty regarding the outcomes of the vital sign features, as well as the vital sign feature qualities. It could also be interpreted as an evaluation of the information carried by the vital signs $V$.

For the vital signs stated in Section 2.4, the Shannon entropy was calculated. We also calculated the mean and variance for each vital sign $V$. Table 2 summarizes the feature generation for vital signs $V$. Let $V_{general}$ denote the mean and variance of vital signs $V$ and $V_{entropy}$ denote the entropy of vital signs of $V$; then, $V_{merge}$ is the combination of both $V_{general}$ and $V_{entropy}$.

**Table 2.** Feature Generation.

| Variable | Type | Features |
|---|---|---|
| | Vital sign | SBP, DBP, Pulse Pressure, Oximetry, |
| Vital signs | Mean | Respiratory Rate, Pulse Rate, Body Temperature |
| | Variance | |

### 2.5.3. Missing Value Imputation

Table 3 shows the average portion of missing data in our data set. While the entropy measures data quality based on probability, the amount of missing data results in missing data in the entropy features. In this step, we focus on the Missing at random (MAR) type of missing data as the biomedical data of patients may be documented on different frequency. As $h$ is the frequency required to compute a single $H_q(V)$, if a patient has less data points for that given feature time, we will obtain a *miss* in the entropy feature. Thus, also we considered conducting missing value imputation on entropy features. Please note that we did not compute entropy-based features on imputed vital sign data, but only on the original data. This is how we made sure that we measured the data quality of the original data, not the data after imputation.

**Table 3.** Missing data analysis in vital signs.

| | | | | Missing Proportion Mean(%) | | | |
|---|---|---|---|---|---|---|---|
| | **SBP** | **DBP** | **Pulse Pressure** | **Oximetry** | **Respiratory Rate** | **Pulse Rate** | **Temperature** |
| Entropy | 0.23(0.07) | 0.23(0.07) | 0.24(0.07) | 11.38(0.58) | 0.83(0.11) | 0.11(0.04) | 7.76(0.31) |
| Mean | 0 | 0 | 0 | 0 | 0 | 0 | 0 |
| Variance | 0 | 0 | 0 | 0 | 0 | 0 | 0 |
| Merge | 0.08(0.12) | 0.08(0.12) | 0.08(0.12) | 3.79(5.41) | 0.28(0.4) | 0.04(0.06) | 2.59(3.69) |

For vital signs features, the median was used for the entropy $V_{entropy}$ and variance. We defined a normal range for each vital signs feature, through the advice of clinicians. The missing value imputation method for the mean and other vital signs raw data were random imputations from the defined normal range, as shown in Table 4.

**Table 4.** Missing Value Imputation.

| Variable | Imputation Method | Variable |
|---|---|---|
| **Vital Signs** | Median | Shannon entropy, Variance |
| | Random imputation from normal range | Mean SBP, DBP, Pulse pressure, Oximetry, Respiratory rate, Pulse rate, Body temperature |

## 3. Results

This section presents the results and analysis of the cohort, setting time, time gap variation, and feature window variation with the proposed entropy-based framework.

### 3.1. Cohort Analysis

The final cohort consisted of 4278 patients, of which 1631 had AKI onset during their ICU stay and labeled as AKI patients, while 2647 patients did not have AKI occurrence and were labeled as non-AKI patients. The training, validation, and testing data rates were set to 80%, 10%, and 10%, respectively. The training and validation cohort included 1492 AKI patients and 2382 non-AKI patients, while the testing cohort included 139 AKI patients and 625 non-AKI patients.

Table 5 shows a detailed comparison of the AKI and non-AKI patients cohort analysis, in terms of their demographics. It showed a statistical significant differences ($p < 0.01$) on time span in ICU in days, age and BMI between AKI and non-AKI group. AKI group tend to stay longer in the ICU, older in age, and had lower BMI value.

An aggregate analysis of vital sign and vasopressor medication between AKI and non-AKI cohort are shown in Table 6. In the comparison, the AKI group had a significant difference ($p < 0.01$) in vital signs, compared to the non-AKI group, except for pulse pressure on the training and validation cohort. In the testing cohort, only respiratory rate and pulse rate showed significant difference ($p < 0.01$). There were significant difference ($p < 0.01$) in the vasopressor medications of vasopressin, Norepinephrine, and Epinephrine between the AKI and non-AKI patients in training and validation sets. There were only significant difference ($p < 0.01$) in Norepinephrine and Epinephrine on the testing set.

In ventilatory support, statistic differences ($p < 0.01$) were shown in Table 7 in the fraction of inspired oxygen (FiO$_2$), positive end-expiratory pressure and continuous positive airway pressure (PEEP/CPAP), mean arterial pressure (MAP), total respiratory rate (RR) between the AKI and non-AKI patients in training and validation sets. Mean airway pressure (Paw), exhaled VT and exhalations volume per time unit(MV) did not show statistic significant differences.

**Table 5.** Cohort Analysis of Patient Population and Demographics.

| Variable | Mean | Training and Validation | | p-Value | Testing | | p-Value |
|---|---|---|---|---|---|---|---|
| | (STD) | AKI | Non-AKI | | AKI | Non-AKI | |
| **Patient population, N** | 4278 | 1492 | 2382 | – | 139 | 265 | – |
| Time span in ICU(days) | 11.29 (12.5) | 17.55 (17.42) | 7.4 (5.95) | $p < 0.01$ ** | 16.02 (10.61) | 8.43 (7.85) | $p < 0.01$ ** |
| **Demographic** | | | | | | | |
| Age | 60.61 (16.47) | 63.78 (16.5) | 58.68 (16.17) | $p < 0.01$ ** | 63.52 (16.93) | 58.54 (15.88) | $p < 0.01$ ** |
| BMI | 23.55 (4.95) | 23.04 (4.89) | 23.92 (4.9) | $p < 0.01$ ** | 23.2 (4.1) | 23.37 (5.81) | $p < 0.01$ ** |
| Male | 2806 (65.59%) | 975 (65.35%) | 1574 (66.08%) | 0.64 | 91 (65.47%) | 166 (62.64%) | 0.58 |
| Female | 1472 (34.41%) | 517 (34.65%) | 808 (33.92%) | 0.64 | 48 (34.53%) | 99 (37.36%) | 0.58 |

*: $p < 0.05$ ; **: $p < 0.01$.

**Table 6.** Cohort Analysis of Vital Signs and Medications.

| Variable | Training and Validation | | | | Testing | | |
|---|---|---|---|---|---|---|---|
| Mean (STD) | All | AKI | Non-AKI | p-Value | AKI | Non-AKI | p-Value |
| **Vital signs, N** | 4278 | 1492 | 2382 | — | 139 | 265 | — |
| SBP(mmHg) | 131.4 (23.68) | 129.69 (25.04) | 132.41 (22.68) | $p < 0.01$ ** | 130.75 (26.76) | 132.23 (22.48) | 0.56 |
| DBP(mmHg) | 78 (16.77) | 76.51 (17.43) | 78.85 (16.26) | $p < 0.01$ ** | 76.15 (18.59) | 79.59 (15.79) | 0.05 |
| Pulse pressure | 53.35 (19.42) | 53.18 (20.63) | 53.49 (18.55) | 0.62 | 54.6 (23.38) | 52.42 (17.72) | 0.29 |
| Oximetry(%) | 98.21 (2.87) | 97.93 (3.45) | 98.39 (2.37) | $p < 0.01$ ** | 98.09 (3.51) | 98.3 (2.92) | 0.53 |
| Respiratory rate | 18.86 (3.92) | 19.42 (4.36) | 18.44 (3.44) | $p < 0.01$ ** | 20.54 (5.74) | 18.6 (3.49) | $p < 0.01$ ** |
| Pulse rate(/min) | 90.17 (19.98) | 94.48 (21.12) | 87.45 (18.68) | $p < 0.01$ ** | 95.22 (21.37) | 87.68 (19.24) | $p < 0.01$ ** |
| Temperature(Celsius) | 36.63 (0.93) | 94.48 (21.12) | 36.58 (0.87) | $p < 0.01$ ** | 36.76 (1.11) | 36.57 (0.82) | 0.05 |
| **Medication, Vasopressors** | | | | | | | |
| Vasopressin | 24 (0.56%) | 13 (0.87%) | 7 (0.29%) | $p < 0.05$ * | 1 (0.72%) | 3 (1.13%) | 0.69 |
| Norepinephrine | 826 (19.31%) | 398 (26.68%) | 348 (14.61%) | $p < 0.01$ ** | 38 (27.34%) | 42 (15.85%) | $p < 0.01$ ** |
| Dopamine | 451 (10.54%) | 156 (10.46%) | 253 (10.62%) | 0.87 | 14 (10.07%) | 28 (10.57%) | 0.88 |
| Epinephrine | 278 (6.5%) | 145 (9.72%) | 102 (4.28%) | $p < 0.01$ ** | 16 (11.51%) | 15 (5.66%) | $p < 0.05$ * |
| Dobutamine | 59 (1.38%) | 23 (1.54%) | 31 (1.3%) | 0.54 | 2 (1.44%) | 3 (1.13%) | 0.79 |

*: $p < 0.05$; **: $p < 0.01$.

**Table 7.** Cohort Analysis of Ventilatory Support.

| Variable | Training and Validation | | | | Testing | | |
|---|---|---|---|---|---|---|---|
| Mean (STD) | All | AKI | Non-AKI | p-Value | AKI | Non-AKI | p-Value |
| **Ventilatory Support** | 4287 | 1492 | 2382 | – | 139 | 265 | – |
| $FIO_2$ | 70.95 (25.87) | 75.4 (26.07) | 67.7 (25.27) | $p < 0.01$ ** | 78.62 (25.33) | 65.89 (24.73) | $p < 0.01$ ** |
| PEEPCPAP | 4.87 (1.69) | 5.06 (1.78) | 4.71 (1.59) | $p < 0.01$ ** | 5.09 (1.72) | 4.86 (1.71) | 0.24 |
| PAW | 23.13 (6.73) | 23.21 (6.72) | 22.89 (6.63) | 0.61 | 23.76 (8.1) | 24.38 (6.71) | 0.8 |
| MAPS | 11.62 (2.55) | 11.93 (2.65) | 11.34 (2.52) | $p < 0.05$ * | 10.97 (1.47) | 11.47 (2.06) | 0.47 |
| TOTRR | 18.89 (5.36) | 19.76 (5.8) | 18.1 (4.81) | $p < 0.01$ ** | 19.01 (5.31) | 17.71 (4.74) | 0.44 |
| VTEXH | 0.52 (0.11) | 0.52 (0.11) | 0.53 (0.11) | 0.28 | 0.51 (0.11) | 0.49 (0.13) | 0.63 |
| MVEXH | 9.57 (2.8) | 9.79 (3.03) | 9.38 (2.61) | 0.13 | 9.72 (2.7) | 9.08 (2.16) | 0.44 |

*: $p < 0.05$ ; **: $p < 0.01$.

### 3.2. Setting Time and Missing Data Analysis Results

The setting time $T_{ST}$ is the minimum time interval between a patient's admission time and their first vital sign data entry. Table 8 shows the mean setting times of Pulse pressure (SBP, DBP), Oximetry, Respiratory rate, Pulse rate, and Temperature. As the average setting time in most vital signs was around 3 h $T_{ST} = 3$, we set 3 h as our setting time in this research. Figure 4 visualizes the averaged setting time between patient's first vital sign data entry and ICU admission. In other words, if the time gap of the prediction task was less than 3 h, a lot of missing data may occur, thus affecting the model performance. In another word, we only consider data collected after $T_{ST} = 3$ h of the patient's admission time.

**Table 8.** Mean setting time for vital signs.

| Vital Signs | Mean Setting Time (h) |
|---|---|
| Pulse pressure (SBP, DBP) | 2.93 |
| Oximetry | 8.39 |
| Respiratory rate | 3.00 |
| Pulse rate | 2.99 |
| Temperature | 3.05 |

**Figure 4.** Visualization plot for setting time between different vital signs.

### 3.3. Classification and Evaluation Criteria

Random Forest (RF) machine learning algorithm [12] was employed to compare results between $V_{entropy}$, $V_{merge}$, and $V_{general}$, where the time gap of AKI onset to 48 h before AKI onset and a feature window of 24 h was used. After finding the critical point in the time gap, we used it in the feature window variation task, studying feature windows of 24 h to 48 h with a time gap of 24 h.

The RF model can predict whether a patient will have AKI onset or not during ICU admission, by providing a probability. The probability is determined by the ratio of the decision trees that give positive results in the total number of trees. In our RF model, we set the number of multiple decision trees as 300 and the number of splits to 21. Ten-fold validation was used.

The performance was evaluated using the accuracy, area under the ROC curve (AUROC), and net reclassification improvement (NRI) metrics. The accuracy measures the percentage of correctly classified samples of the total samples under a given threshold. The AUROC evaluates the overall model performance. The NRI aims to provide an objective method to quantify improvements in categories in models [18]. It measures how well a new model correctly reclassifies subjects that were not correctly classified in the old model. In our case, the new model was the model with increment in time $t$, while the old model for comparison was the model with prior time (i.e., $t - 1$). F value from F test was conducted in order to compare the variance between two groups.



### 3.4. Classification Performance with Time Gap Variation

In this section, we reveal the time gap variation $T_{TP}$ results. We studied the time gap from AKI onset to 48 h $T_{TP} = [0, 48]$ before onset using a feature window size of 24 h $T_{FW} = 24$. This is critical for identifying the most important point of time between slight AKI signs and emerging AKI oscillation. We compared the accuracy, AUROC, and NRI performance of $V_{entropy}$, $V_{mean}$, $V_{variance}$, and $V_{merge}$. The Model performance results are shown in Figure 5. It was shown in AUROC that the $V_{merge}$, $V_{mean}$ and $V_{variance}$, performed steadily with different time gap variation. $V_{entropy}$ performed better when time gap is longer.

Figure 6 shows the F value from F test of AUROC along with the time gap. The F value on AUROC showed that the peak in F value was at 30 h; namely 30 h was the point that contributed the most information and was the turning point from slight AKI signs to emerging AKI oscillation. Therefore, clinicians may consider the patient has higher risk of having AKI onset. The overall model performance increased drastically, when starting at this point in time. The trend was most obvious in NRI for both $V_{entropy}$ and $V_{merge}$. As the NRI seeks to quantify whether a new marker provides clinically relevant improvements in model prediction, this result indicates that when passing the critical point of 30 h, the shorter the time gap and the more information available, the better the models performed.

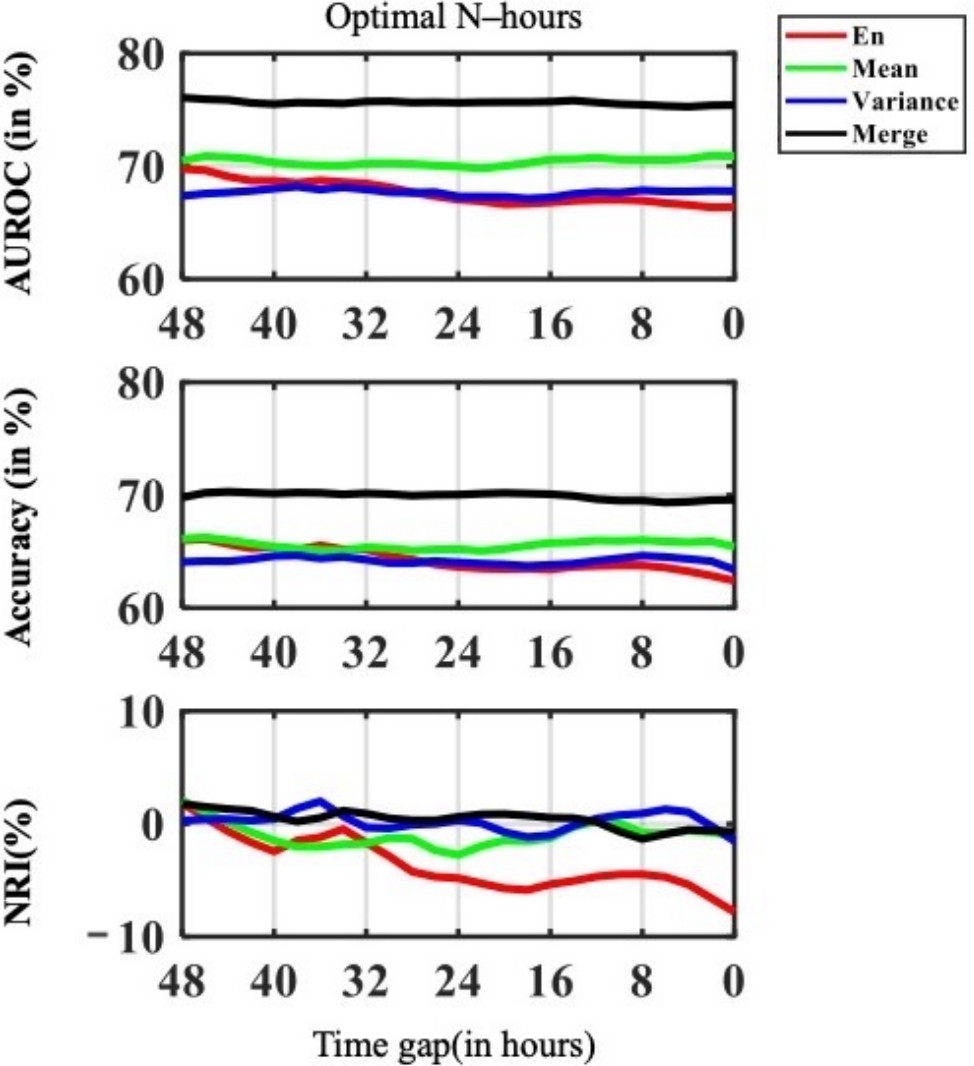

**Figure 5.** Visualization plot for time gap variance on AUROC, Accuracy and NRI performance.

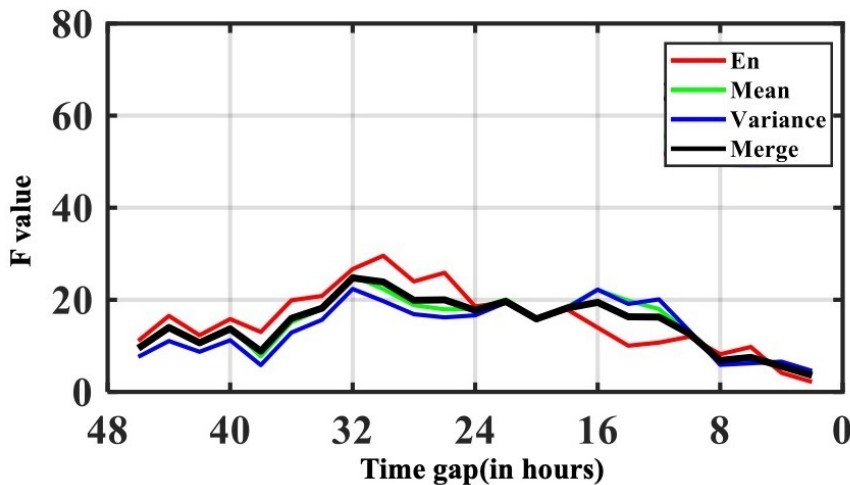

**Figure 6.** Visualization plot for time gap variance F value from F test on AUROC.

As 30 h was the critical point of time between slight AKI signs and rapid AKI oscillation, as shown in in Tables 9 and 10 , we conducted a comparison between using a time gap of less than 30 h (after critical point, rapid AKI oscillation) and a time gap of more than 30 h (before critical point, slight AKI signs). Statistically significant differences ($p < 0.01$) were observed in and $V_{entropy}$ in AUROC; $V_{entropy}$ and $V_{merge}$ inaccuracy, NRI and precision. $V_{entropy}$ and $V_{mean}$ in Recall

**Table 9.** Comparison between using a time gap of less than 30 h (rapid AKI oscillation) and a time gap greater than 30 h (slight AKI signs) on AUROC, Accuracy and NRI.

| Feature Types | AUROC Mean (STD) | | | Accuracy(%) Mean (STD) | | | NRI(%) Mean (STD) | | |
|---|---|---|---|---|---|---|---|---|---|
| Time Gap | Before Critical Point | After Critical point | *p* Value | Before Critical Point | After Critical Point | *p* Value | Before Critical Point | After Critical Point | *p* Value |
| Entropy | 69.09 (0.82) | 67.36 (0.99) | $p < 0.01$ ** | 65.66 (0.80) | 64.02 (1.08) | $p < 0.01$ ** | −0.9 (1.86) | −4.19 (2.27) | $p < 0.01$ ** |
| Mean | 70.81 (0.31) | 70.35 (0.53) | 0.07 | 66.01 (0.42) | 65.48 (0.55) | 0.06 | −0.22 (1.09) | −1.29 (1.10) | 0.09 |
| Variance | 67.68 (0.49) | 67.67 (0.56) | 0.96 | 64.13 (0.31) | 64.18 (0.60) | 0.79 | 0.19 (0.85) | 0.17 (1.23) | 0.95 |
| Merge | 75.87 (0.30) | 75.58 (0.34) | 0.09 | 70.3 (0.31) | 69.89 (0.46) | $p < 0.05$ * | 1.3 (0.30) | 0.2 (0.95) | $p < 0.05$ * |

*: $p < 0.05$; **: $p < 0.01$.

**Table 10.** Comparison between using a time gap of less than 30 h (rapid AKI oscillation) and a time gap greater than 30 h (slight AKI signs) on Recall, and Precision.

| Feature Types | AUROC Mean (STD) | | | Accuracy(%) Mean (STD) | | |
|---|---|---|---|---|---|---|
| Time Gap | Before Critical Point | After Critical Point | *p* Value | Before Critical Point | After Critical Point | *p* Value |
| Shannon entropy | 55.74 (0.89) | 53.88 (1.44) | $p < 0.01$ ** | 60.57 (1.04) | 58.42 (1.44) | $p < 0.01$ ** |
| Mean | 60.82 (1.37) | 59.49 (0.99) | $p < 0.05$ * | 59.98 (0.46) | 59.51 (0.69) | 0.19 |
| Variance | 55.72 (1.24) | 55.1 (0.94) | 0.25 | 58.31 (0.35) | 58.45 (0.82) | 0.65 |
| Merge | 66.76 (0.67) | 66.33 (0.99) | 0.4 | 64.7 (0.44) | 64.18 (0.58) | $p < 0.05$ * |

*: $p < 0.05$; **: $p < 0.01$.

### 3.5. Classification Performance with Feature Window Variation

For feature window variation, we varied the feature window size from 24 h to 48 h before onset $T_{FW} = [24, 48]$, with a time gap of 24 h $T_{TP} = 24$. The feature window variation provided insights, in terms of clinical practice, about how much data is required for the best prediction performance and how different features react differently to different time windows. The model performance in the feature variation task is shown in Figure 7. A comparison of 24 h and 48 h feature windows is shown in Tables 11 and 12.

The performance of $V_{merge}$ was the highest among all the other features, such as $V_{mean}$, $V_{variance}$, and $V_{entropy}$. Both the AUROC accuracy and NRI of $V_{merge}$ consistently improved with increasing feature data availability. The longer the feature window, the more data was available for the model to learn from. Model performance with $V_{entropy}$ outperformed $V_{mean}$ when the feature window size was bigger than 28 h. The overall model performance of $V_{entropy}$ improved drastically with more data availability, as shown in Figure 7.

In terms of accuracy, we see a merging trend in performance of the proposed $V_{entropy}$ feature. We can reasonably conclude that with fewer features available, our proposed entropy-based feature $V_{entropy}$ performed better. While there the accuracy was steady in $V_{mean}$, and there is no evidence that more data in $V_{variance}$ would make the prediction better.

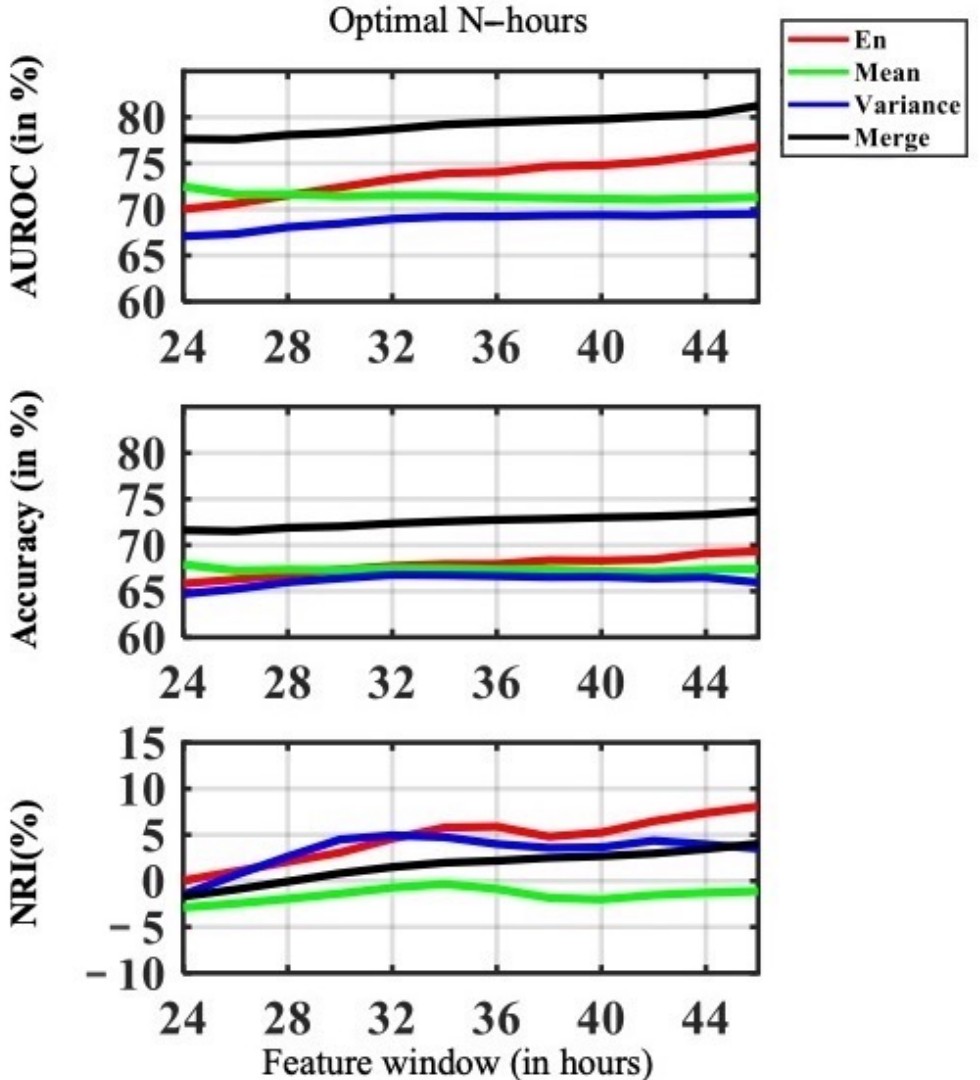

**Figure 7.** Visualization plot for feature window variation performance, with the feature window size varying from 24 h to 48 h before onset, with a time gap of 24 h.

In terms of NRI, $V_{entropy}$ was the highest among other features after 32 h. This implied that the increase in feature window had a high impact on $V_{entropy}$ model performance, such that the $V_{variance}$ performance was more sensitive to the length of feature window and data availability. The model performance of $V_{entropy}$ was sensitive to the feature window size of 24 h to 36 and 38 to 48 h of feature window size. $V_{mean}$ was not sensitive to feature window variance. Similar behavior was found in $V_{variance}$, the amount of available data was critical to $V_{variance}$ between 24 to 30 h of feature window size In comparison, despite the fact that $V_{entropy}$ performed well, in terms of accuracy, we drew the same conclusion as for AUROC; that is, $V_{entropy}$ improved drastically with more data and a longer feature window. The NIR showed that $V_{entropy}$ is steadily sensitive to data availability.

In Tables 11 and 12 we showed an improvement comparison between feature window size of 24 h $T_{FW} = 24$ and 48 h $T_{FW} = 48$. The $V_{entropy}$ feature had the highest performance improvement in AUROC, NRI, Recall and Precision. At the same time, more data may not improve the performance of $V_{mean}$. Compared to $V_{variance}$, more data available improves $V_{merge}$'s performance across all metrics.

**Table 11.** Comparison of 24 h and 48 h feature windows on AUROC, Accuracy, NRI.

| Feature Types | AUROC (%) | | | Accuracy (%) | | | NRI (%) | | |
|---|---|---|---|---|---|---|---|---|---|
| Feature Window | 24 h | 48 h | Improvement | 24 h | 48 h | Improvement | 24 h | 48 h | Improvement |
| Shannon entropy | 70.01 | 76.8 | 6.79 | 65.82 | 69.33 | 3.5 | – | 8.32 | 8.32 |
| Mean | 72.48 | 71.31 | −1.16 | 67.91 | 67.42 | −0.48 | – | −0.88 | −0.88 |
| Variance | 67.08 | 69.51 | 2.42 | 64.7 | 65.96 | 1.26 | – | 3 | 3 |
| Merge | 77.64 | 81.24 | 3.59 | 71.62 | 73.67 | 2.05 | – | 4.4 | 4.4 |

**Table 12.** Comparison of 24 h and 48 h feature windows on Precision and Recall.

| Feature Types | Recall(%) | | | F1-Score | | | NRI (%) | | |
|---|---|---|---|---|---|---|---|---|---|
| Feature Window | 24 h | 48 h | Improvement | 24 h | 48 h | Improvement | 24 h | 48 h | Improvement |
| Shannon entropy | 54.55 | 64.59 | 10.03 | 55.37 | 59.32 | 3.94 | 54.96 | 61.84 | 6.88 |
| Mean | 60.56 | 59.28 | −1.27 | 57.56 | 57.48 | −0.08 | 59.02 | 58.37 | −0.65 |
| Variance | 52.2 | 54.97 | 2.76 | 53.97 | 55.9 | 1.92 | 53.07 | 55.43 | 2.35 |
| Merge | 67.43 | 70.88 | 3.45 | 61.48 | 64.37 | 2.52 | 64.51 | 67.47 | 2.95 |

## 4. Discussion and Limitation

In this study, we proposed a novel entropy-based feature engineering framework for vital signs, based on their frequency of records and clinical availability. Both quantitative analysis of the features and data quality are taken into analysis. In addition, we conducted feature window and time gap experiment in order to determine the best time window to maximize the accuracy of AKI prediction in ICU.

The etiology of AKI is usually multi-factorial. Dehydration, infection, renal toxic medications, and contrast medium injection during computed tomography scanning are common risk factors for AKI [19]. Clinicians are usually aware of AKI only when patients' urine output decrease or elevated serum creatinine occur. In 2014, the FDA approved a commercial urine stress biomarker, Nephrocheck®, to be used for ICU patients for early prediction of AKI with 12-h time gap windows in advance [20]. However, the commercial kit is very expensive and has not been approved by other countries other than the United States. On the other hand, we sought to integrate an AKI prediction algorithm into electronic health information system in our hospital to provide a real time AKI risk probability for ICU patients without spending additional cost and nursing labors. To achieve this goal, we mathematically define the EHR data availability according to different time gaps and feature windows to apply state-of-the-art machine learning techniques.

Compared to recent deep learning-based AKI prediction studies [6,21,22], we not only conducted AKI prediction, but also proposed an overall framework that covers feature pre-processing, missing value imputation, and proposed a novel entropy-based vital sign feature engineering. Moreover, we conducted a fine-grained analysis considering both time gap variation and feature window variation. In feature window examination, we tested the feature window length from 24 h to 48 h before AKI onset, with a fixed time gap of 24 h. Our results demonstrated that early AKI can be predicted, based on our proposed vital sign entropy-based features, with a feature window of 24 h and a time gap of 24 h. The best-performing model was obtained when using all available features with the longest feature window. The result implies that both the frequency of clinical records and data quality are important for AKI prediction model.

Our study has several limitations. First, our cohort is derived from a single hospital, and the prediction algorithm has not been externally validated with other cohorts which may limit its generalization. Second, the best feature window length and time gap length may vary depending on the availability of the clinical records. However, our work could still provide a practical data mining process for developing a prediction model in ICU.

## 5. Conclusions

We introduced an end-to-end practical framework from missing data handling, entropy-based feature engineering, to different time gap and feature window length analysis in an ICU data set for AKI prediction. In the missing data handling, our proposed framework is able to address both the missing at random (MAR) and missing not at random (MNAR) types of missing data in clinical practice. As RF algorithms are widely used in various settings, our proposed method provides the practical missing data handling that will improve model performance. We studied the relationships among different time gap variations and feature window variations with the proposed vital sign entropy feature for AKI prediction. This work could provide a guidance for feature windows selection and missing data processing during the development of a prediction model in ICU.

**Author Contributions:** Conceptualization: C.-T.H. (Chun-Te Huang), C.-H.C. and C.-M.L.; Methodology: T.-J.W., L.-C.C. and R.-K.S.; Software: Y.-L.T.; Data Curation: C.-C.H. and M.-S.W.; Writing: R.-C.C., C.-T.H. (Chun-Te Huang) and C.-M.L.; Investigation: C.-L.W.; Validation: C.-T.H. (Chia-Tien Hsu) and K.-C.P. All authors have read and agreed to the published version of the manuscript.

**Funding:** This research was funded by Ministry of Science and Technology in Taiwan under grant number MOST 109-2321-B-075A-001-1.

**Institutional Review Board Statement:** The study was conducted according to the guidelines of the Declaration of Helsinki, and approved by Institutional Review Board of Taichung Veterans General Hospital under case number SE20249B, approved on 7 May 2020.

**Informed Consent Statement:** Patient consent was waived, due to retrospective study and all data had been deidentified under review by the Institutional Review Review Board of Taichung Veterans General Hospital.

**Data Availability Statement:** The data that support the findings of this study are available upon request from the corresponding author. The data are not publicly available, due to the constraint of Taiwanese Personal Information Protection Act.

**Conflicts of Interest:** The authors declare no conflict of interest. The funders had no role in the design of the study; in the collection, analyses, or interpretation of data; in the writing of the manuscript, or in the decision to publish the results.

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
