# Peer review of "Entropy-Based Time Window Features Extraction for Machine Learning to Predict Acute Kidney Injury in ICU"

_applsci, doi:10.3390/app11146364_

Round 1
Reviewer 1 Report
Thank you for the study and having me review this manuscript. In brief this is a highly interesting article. The authors rightly pointed out the issues with health informatics and predictive analytics, which is the variable nature of how the feature window and time span (prediction window) are defined in various studies, as well as how best to handle missing data. Variety of approaches are tried and tested in literature including substituting missing as null or imputing mean or median with no clear advantages or disadvantages on the final outcome. Therefore, using the element of entropy to measure data quality in account of missing values in any feature time, adds value to this paper.
(Please note that “controls” refer to the non-AKI patients, in the below discussion)
Major comments
- I can understand that AKI cases were selected based on an AKI onset beyond 72 hrs of ICU admission, in order to have the sufficient feature window and time span for analysis. However, the controls are not selected based on a minimum 72 hours of ICU stay, and the feature windows are of random 24 hrs, with no corresponding time span. In my research we would commonly adopt equivalent entry criteria for both cases and non-AKI controls. i.e. Exclude entirely patients who had less than 72 hours ICU stay, in order to ensure comparable feature window and time span for AKIs and controls. By including short stay controls (i.e. ICU stay under 72 hrs), the authors may instead include very sick patients who passed on due to reasons entirely unrelated to kidney disease, or include controls with limited features for analysis. In that regard, I am also unclear, how did the authors select controls with 48 hrs of feature window? Did we then exclude the controls with feature time under 48 hours? The authors should respond by discussing the pros and cons of their methodology versus having controls of similar feature window and time span. Please keep to the final word limit.
- Section 3.1: First paragraph: If the training cohort has 1306 AKI patients, how is it the validation and testing cohorts have another 1631 AKI patients? The numbers of AKI and controls do not add up (see Figure 5). Do the authors mean the training and validation cohorts are as one? And the derivation and 10-fold validation were performed in this same training + validation cohort? Please clarify this, and suggest show the absolute numbers and breakdown in section 3.1 + Figure 5.
- As this is an ICU study, under Figure 6 – as a clinician myself, we cannot interpret blood pressure, and respiratory rate, and oximetry, without knowing if these patients are on vasoactive drugs and/or mechanical ventilation. Please include these 2 details in Figure 6. You need not add these indices into the analytics, as that would be a massive revision, but please note, that patients on vasoactive drugs and ventilatory support will naturally have more vital sign parameters done and these are sicker patients who will more likely get kidney injury, and these influence the entropy probability too. The authors should review these and comment as necessary.
- Classification performance. Ideally, in addition to AUROC and accuracy, for such predictive models, we should know the recall and precision. A model with good AUROC or balanced accuracy, may not necessarily score well in recall and precision; the latter are more important for AKI prediction in clinical practice. I would like to see these details in figure 8 and 11. At the very least, the recall, please.
Minor note:
- Standardise descriptive: If “time gap” is used to define the prediction window, then do not use this term to describe other time windows (e.g. in the definition of setting time, it is described as the time gap between admission and first vital sign. This confuses the reader as “time gap” was specifically meant to describe the prediction window).
- The entropy concept is difficult to grasp. I am not sure how best, but if the authors could describe entropy in more lay language to benefit the non-computer science readers.
Thank you.
Author Response
Thank you for the study and having me review this manuscript. In brief this is a highly interesting article. The authors rightly pointed out the issues with health informatics and predictive analytics, which is the variable nature of how the feature window and time span (prediction window) are defined in various studies, as well as how best to handle missing data. Variety of approaches are tried and tested in literature including substituting missing as null or imputing mean or median with no clear advantages or disadvantages on the final outcome. Therefore, using the element of entropy to measure data quality in account of missing values in any feature time, adds value to this paper.
Thank you so much for the advice. We will address your suggestions as follows.
- I can understand that AKI cases were selected based on an AKI onset beyond 72 hrs of ICU admission, in order to have the sufficient feature window and time span for analysis. However, the controls are not selected based on a minimum 72 hours of ICU stay, and the feature windows are of random 24 hrs, with no corresponding time span. In my research we would commonly adopt equivalent entry criteria for both cases and non-AKI controls. i.e. Exclude entirely patients who had less than 72 hours ICU stay, in order to ensure comparable feature window and time span for AKIs and controls. By including short stay controls (i.e. ICU stay under 72 hrs), the authors may instead include very sick patients who passed on due to reasons entirely unrelated to kidney disease, or include controls with limited features for analysis. In that regard, I am also unclear, how did the authors select controls with 48 hrs of feature window? Did we then exclude the controls with feature time under 48 hours? The authors should respond by discussing the pros and cons of their methodology versus having controls of similar feature window and time span. Please keep to the final word limit.
Thank you for such an important suggestion. In this version, we re-define non-AKI cohort a by excluding patients who had less than 72 hours ICU stay, to make sure AKI and non-AKI cohort features were both used 72 hours after ICU admission. Non-AKI cohort changes from 6067 to 2674 <Line 117> and we accordingly report the updated result <Line 107 to 118>, this makes the classification task data more balanced.
As data collection process, under TVGH ICU policy, after ICU admission, normally a patient vital signs will be recorded every 2 hours, but not in general ward. Actually, the core research question for the work is to discuss the prediction results on variable feature window <24-48 hours> and time gap <We also add three notations on Table 1 (TTP, TST, TFW)>. The pros and cons have been added <Line 295-297>. Thank you for the advice.
2. Section 3.1: First paragraph: If the training cohort has 1306 AKI patients, how is it the validation and testing cohorts have another 1631 AKI patients? The numbers of AKI and controls do not add up (see Figure 5). Do the authors mean the training and validation cohorts are as one? And the derivation and 10-fold validation were performed in this same training + validation cohort? Please clarify this, and suggest show the absolute numbers and breakdown in section 3.1 + Figure 5.
Thank you for the clarification. We have updated the correct numbers <Figure 1 flow, Line 228-234, Figure 5, 6>. We have checked the sum is correct, thanks for the careful reviews.
3.As this is an ICU study, under Figure 6 – as a clinician myself, we cannot interpret blood pressure, and respiratory rate, and oximetry, without knowing if these patients are on vasoactive drugs and/or mechanical ventilation. Please include these 2 details in Figure 6. You need not add these indices into the analytics, as that would be a massive revision, but please note, that patients on vasoactive drugs and ventilatory support will naturally have more vital sign parameters done and these are sicker patients who will more likely get kidney injury, and these influence the entropy probability too. The authors should review these and comment as necessary.
Thank you so much for the advice. We have added vasoactive drugs and ventilatory support data <Figure 6, 7> and the corresponding explain <Line 242-257>
4.Classification performance. Ideally, in addition to AUROC and accuracy, for such predictive models, we should know the recall and precision. A model with good AUROC or balanced accuracy, may not necessarily score well in recall and precision; the latter are more important for AKI prediction in clinical practice. I would like to see these details in figure 8 and 11. At the very least, the recall, please.
Thank you so much for the advice. We have added the precision and recall as Figure 15. The explain was also offered <Line 351-355>.
5.Standardise descriptive: If “time gap” is used to define the prediction window, then do not use this term to describe other time windows (e.g. in the definition of setting time, it is described as the time gap between admission and first vital sign. This confuses the reader as “time gap” was specifically meant to describe the prediction window).
Thanks for the clarification. We have presented the updated notation on Table 1. Also we modify the Figure 1 to clearly differentiate those three terms: (1) Feature windows (24 to 48 hrs in this work) (2): Time gap (0-48 hrs in this work). <Line 319> <Line 351> also describe the variable in more detail.
6.The entropy concept is difficult to grasp. I am not sure how best, but if the authors could describe entropy in more lay language to benefit the non-computer science readers.
Thanks for the suggestion. Entropy can help to capture another concept compared to mean, variance and slope. Some related work has been addressed [Line 59-70], presenting how entropy has been applied in clinicals. Also we add one more line in [Line 201].
Thank you for the careful review and suggestion. We also made another round editing to modify some typos and grammars.

Reviewer 2 Report
the well work performed by the authors it's very interesting. it well know that Acute Renal injury can lead to a rapid worsening of the patient's condition if not diagnosed or treated in time. The aim of the author work is to design a new predictive model that can help clinicals to diagnose AKI. From my point of view the designed method by authors strongly correlates with data quality presented by them in the paper. The number of features, patients and vital signs used to test is well suitable, although it would be interesting to extend the cohort in the future, to improve the accuracy and reproducibility of the proposed model as well as include new time gaps in the model which is crucial for AKI development. In my opinion the work is a good starting point for future implementations in helping clinicals.
Author Response
Thank you so much for the advice. Actually the system has been applied and we keep collected the feedback from clinicians to validate the efficiency. The suggestion has been added as a part of limitation <Line 398-405>. We also made another round editing to modify some typos and grammars here and there. Thank you so much.

Reviewer 3 Report
Review
Dear Editor
I read with interest the publication “Entropy-Based Time Window Features Extraction For Machine Learning To Identify Acute Kidney Injury”.
Current data analysis-data engineering capabilities are increasingly enabling the process of patient diagnosis and therapy.
But the ability to analyze extremely large and diverse structured and unstructured data is becoming a data science problem. Whether the aim is to present complex ways of mathematical analysis of data, whether the conclusions and the solutions obtained do not become a by-product.
Turning to the details and analysis of the presented publication, I literally just have the impression that it exalts itself with mathematical possibilities, while the analysed material is an additional element.
Intervening 30 hours before AKI develops is looking into a glass ball.
"We suggested clinicians intervene before this point in time."[314-317]
The authors do not understand the clinical situation where each patient is treated individually and the concept of being 30 hours ahead of the clinical situation is irrational.
The initial hours are "golden hours" as early prediction and identification of AKI could help clinicians to give timely intervention and avoid the grave prognosis of end-stage renal disease. [317,318].
Once a patient is admitted to care, any moment of organ dysfunction, decreased diuresis, potassium elevation, circulatory dysfunction, even before the development of permanent organ damage is an indication for therapy.
Another problem in the treatment of patients is the highly individualized and personalized clinic - this also applies to haemodynamic parameters and their translation into renal function
“Our results suggested that 30 hours before AKI onset is the tipping point between slight AKI omen and rapid AKI oscillation. We suggested clinicians consider patients at high risk of developing AKI onset at this critical point of time”.[352-353]
I find no clinical benefit from the observations presented. On the other hand, I would not like to see the concept of 30 golden hours for decision time and treatment until permanent dysfunction occurs in front of every patient treated. Patients often do not have so much time.
Author Response
Dear reviewer 3,
Thank you for the suggestions.
Please kindly check the attachment.

Round 2
Reviewer 3 Report
Dear Editor
I have once again made my way through the reviewed work. Unfortunately for me as a medical professional it is painful to read the clinical references describing the symptoms of renal failure: "Impaired renal function can lead to anemia, mineral-bone disease, and general edema "[23-24], the aetiology of AKI : "Dehydration, sepsis, nephrotoxic medications, and contrast medium injection during computed tomography scaninng are common risk factors for AKI.", and the author's references to the early signs of renal dysfunction "...the early diagnosis of AKI remains a challenge for physicians, given the fact that SCr is a delayed function marker for AKI, and some nephrotoxic antibiotics (e.g., gentamicin) can cause non-oliguric AKI.."? [35-36]. The most interesting, however, is the definition of early renal failure :" Our results suggest that 30 hours before AKI onset is the tipping point of time between slight AKI signs and rapid AKI oscillation. We suggest that clinicians, considering patients at higher high risk of developing AKI onset ..."[419-420]. The publication shows us the possibilities of data science and data analysis, including its supplementation. It is universal and the fact that the authors chose AKI and not heart failure seems a coincidence. Also, the analysis of the data rather foregrounds the fact that patients with AKI had a severe clinical course compared to the others is a conclusion from the observation of clinical parameters. But this is the knowledge we already have. The presented publication has a difficult mathematical message, its clinical conclusions are questionable. I find it difficult to find conviction for the advisability of publishing the paper.
